# Findings and Graduation of Sarcoidosis-Related Uveitis: A Single-Center Study

**DOI:** 10.3390/cells11010089

**Published:** 2021-12-29

**Authors:** Lynn S. zur Bonsen, Dominika Pohlmann, Anne Rübsam, Uwe Pleyer

**Affiliations:** 1Department of Ophthalmology, Charité–Universitätsmedizin Berlin, Augustenburger Platz 1, 13353 Berlin, Germany; dominika.pohlmann@charite.de (D.P.); anne.ruebsam@charite.de (A.R.); 2Berlin Institute of Health, Charité–Universitätsmedizin Berlin, Charitéplatz 1, 10117 Berlin, Germany

**Keywords:** differential diagnosis, IWOS criteria, granulomatosis, neurosarcoidosis, ocular sarcoidosis, uveitis

## Abstract

Ocular involvement is present in up to 79% of sarcoid patients. Uveitis is the main ocular manifestation and presents as a chronic intraocular inflammatory condition with potentially detrimental effects on visual acuity and quality of life. This retrospective study was conducted to explore the incidence and characteristics of ocular sarcoidosis in a single tertiary ophthalmology center. Medical records of 84 patients presenting between June 2007 and March 2021 were analyzed. Based on the “International Workshop on Ocular Sarcoidosis” (IWOS) criteria, ocular sarcoidosis was determined as: definite (*n* = 24; 28.6%), presumed (*n* = 33; 39.3%), probable (*n* = 10; 11.9%), and indefinite (*n* = 17; 20.2%) in our study population. In 43.9% of the definite and presumed cases, the eye was primarily affected. In addition to specific ocular findings, the diagnosis was supported by biopsy (28.6%) and chest x-ray or computer tomography (66.7%). Moreover, an increased soluble interleukin-2 receptor (sIL-2R) expression (76.2%), elevated angiotensin-converting enzyme (ACE) levels (34.8%), and lymphocytopenia (35.1%) were valuable laboratory findings. Co-affected organs were lungs (60.7%), skin (15.5%), and central nervous system (8.3%). Our findings support the prominent role of the eye in the early detection of sarcoidosis. In addition to the IWOS criteria, sIL-2R, in particular, was shown to be relevant in establishing the diagnosis.

## 1. Introduction

Sarcoidosis is a systemic autoimmune disorder based on the presence of non-caseating, epithelioid cell granulomas after infection or when any other cause has been ruled out. The clinical course, organ involvement, and prognosis vary considerably and may explain the delay in diagnosis. Initially, patients often present with unspecific symptoms, such as fatigue or cough. The disease most frequently involves the lungs in the form of bilateral hilar lymphadenopathy (BHL) or pulmonary infiltration [1]. The prevalence of ocular involvement in systemic sarcoidosis has been reported in a range between 6% to 79%, depending on the country with the highest burden of disease in Japan, followed by Northern Europe and the United States [2,3]. In Europe, a north-south gradient can be observed whereby prevalence is twice as high in the North compared to the South. Middle Eastern countries and China exhibit the lowest burden of disease [2].

Notably, in up to one-third of all sarcoidosis patients, ocular findings precede the systemic disease or are key features that lead to its diagnosis [4,5]. All anatomical structures of the eye can potentially be affected by granulomatous changes, although intraocular inflammation in the form of uveitis is the most frequent finding [6,7]. Uveitis causes pronounced discomfort for the patient, including redness, photophobia, epiphora, ocular pain, or a reduction in visual acuity up to blindness in the advanced stage of disease [4]. Based on the “Standardization of Uveitis Nomenclature” (SUN), a classification is made between an anterior, intermediate, and posterior uveitis—depending on the predominantly affected eye segment—with further distinction between an acute or chronic course. In about two-thirds of ocular sarcoidosis patients, the disease has a chronic undulating progression [8]. In general, the disease responds well to steroids, which can be applied locally to the eye or systemically if needed. In some refractory cases, however, the therapeutic options might be expanded to conventional or biologic disease-modifying anti-rheumatic drugs (DMARDS) [9].

Due to the high variability of the disease, the International Workshop on Ocular Sarcoidosis (IWOS) attempted to further classify ocular sarcoidosis in 2009 [10]. In 2018, the diagnostic criteria were revised by this international board of experts and approved [11]. So far, there are only a few reports on the application of these international guidelines. In addition, data on ocular sarcoidosis in Germany are almost lacking. Therefore, we sought to present our institutional experience with the IWOS diagnostic criteria and occurrence of ocular sarcoidosis in this study.

## 2. Materials and Methods

### 2.1. Study Population

This retrospective study was conducted to analyze the occurrence of characteristics of ocular sarcoidosis in a tertiary-care center. Medical records of 84 patients with ocular sarcoidosis diagnosed by a uveitis specialist were collected. Patients with an initial presentation between June 2007 and March 2021 were included. Unspecified cases with a relevant differential diagnosis a priori were not included. This study was performed in accordance with the Declaration of Helsinki.

### 2.2. Ocular Classification

The diagnosis of ocular sarcoidosis was based on the revised IWOS criteria (Table 1). Ocular sarcoidosis was classified in 4 groups as (1) definite, (2) presumed, (3) probable, and (4) indefinite using a combination of intraocular signs and systemic investigations (Table 2). Patients who could not be assigned to any group were designated as indefinite.

Intraocular signs were considered equally positive with either one eye or both eyes of the patients being affected. In addition to the IWOS criteria, an elevated serum soluble interleukin-2 receptor (sIL-2R) level was also evaluated as a positive systemic investigation. The angiotensin-converting enzyme (ACE) serum values of patients taking an ACE inhibitor or systemic immunosuppression were not included in the analysis. For comparability, the individual laboratory values were normalized with respect to the norm ranges of the respective laboratory device since current values from external sources were also accepted. Systemic workup and, especially, imaging of the lungs was performed by the general practitioner or the in-house radiologist. The visual acuity was measured on a decimal scale and converted to Snellen and Logarithm of the Minimum Angle of Resolution (LogMAR) scales. The average of both eyes was calculated and rounded down to the nearest whole number.

The ocular involvement was classified according to the SUN working group [12]. This classification is based on the anatomical focus of uveitis, which was determined by slit lamp examination. Anterior uveitis includes inflammation of the iris and the anterior ciliary body visible as cells and flares in the anterior chamber or the anterior vitreous body, conjunctival injection and iris synechiae. The term intermediate uveitis covers posterior cyclitis, pars planitis, and hyalitis. The focus of the inflammation is found in the vitreous body. Slit lamp examination reveals vitreous haze, vitreous cells, and vitreous opacities up to snowballs. Secondary conditions may include macular edema, vasculitis, or papillitis. These were assessed by multimodality imaging, including fundus photography, optical coherence tomography, fluorescein angiography, as well as indocyanine green angiography. Posterior uveitis describes inflammation of the choroid or retina, which usually occur together due to anatomical proximity. Retinal vasculitis and neuroretinitis are also classified within this group according to the SUN classification. Fundoscopy can reveal retinal as well as choroidal infiltrates, possibly supported by fundus autofluorescence or angiography. Perivascular sheeting, vascular attenuation, cotton-wool spots, or hemorrhage indicate vasculitis. In addition, vitreous affections mentioned in the context of intermediate uveitis may occur with a focus on the posterior pole. Finally, combined involvement of all areas is categorized as panuveitis.

### 2.3. Statistical Analysis

Patient data were obtained from the electronic medical record system and subsequently analyzed using R Statistical Software (version 4.0, Foundation of Computing, Vienna, Austria). Fisher’s exact test was used to compare categorical clinical data. Kruskal–Wallis test and Wilcoxon rank-sum test were used to compare numeric laboratory and demographical data. *p*-values < 0.05 were considered statistically significant.

## 3. Results

### 3.1. Clinical Manifestation

#### 3.1.1. Patient Characteristics

Our 84 patients were clustered into four groups according to the IWOS criteria for the diagnosis of ocular sarcoidosis: definite (*n* = 24; 28.6%), presumed (*n* = 33; 39.3%), probable (*n* = 10; 11.9%), and indefinite (*n* = 17; 20.2%) (Table 3). More than half of the patients were female (50/84, 59.5%), and the mean age of all individuals was 53 years (range 8–87) at the first visit. Females dominated in all groups, except in the indefinite group, where more men were present (58.8%). There was no difference in the average age between diagnostic groups (Kruskal–Wallis test *p* > 0.05). At the time of the first presentation, a diagnosis of sarcoidosis had already been established in almost 40% (31/84) of patients.

Out of 57 patients, 25 (43.9%) with definite or presumed ocular sarcoidosis, namely with positive biopsy or BHL, initially presented with an ocular manifestation. The analysis of visual acuity showed no significant differences between the groups. On average, patients presented with a visual acuity of 6/7.5 (Snellen) or 0.1 LogMAR. Six patients did not achieve Snellen acuity of 6/60 or 1.0 LogMAR with one eye.

#### 3.1.2. Systemic Involvement

The most common co-affected organ in ocular sarcoidosis patients was the lung (*n* = 56; 66.7%). The second most common was skin involvement, which only occurred in the definite and presumed disease groups (in total, 15.5%). Moreover, patients with posterior uveitis (42.9%) showed significantly more skin manifestations as compared to those with intermediate uveitis (11.1%; Fisher’s Exact Test *p* = 0.042).

All other organ manifestations tended to occur in the definite group. Particularly noteworthy was the association of ocular sarcoidosis with neurosarcoidosis (8.3%). Table 3 gives an overview of the organ involvement of sarcoidosis in our patients.

### 3.2. Ocular Findings

#### 3.2.1. IWOS Classification and Uveitis Subtypes

Almost all patients presented with uveitis as ocular involvement. Only one patient showed recurrent scleritis. In total, 31.0% of patients presented with anterior uveitis (26/84), 32.1% with intermediate uveitis (27/84), 16.7% with uveitis posterior (14/84), and 19.0% with panuveitis (16/84). There was a notably higher proportion of intermediate uveitis patients in the probable disease group (60.0%). This result was proven to be statistically significantly higher as compared to the intermediate uveitis group of patients with definite diagnoses (Fisher’s Exact Test *p* = 0.049). Anterior uveitis was most frequently observed in the presumed disease group (36.4%) and posterior uveitis in the definite disease group (29.2%) (Figure 1).

In the context of anterior uveitis, both granulomatous (*n* = 24) and non-granulomatous (*n* = 22) keratic precipitates were observed. Koeppe (*n* = 11) and Busacca (*n* = 1) nodules as iris granulomata were noticed less frequently. The presumed disease group presented with a higher rate of Koeppe nodules (21.2%) compared to the definite disease group (0.0%) (Fisher’s Exact Test *p* = 0.039). Characteristics of intermediate uveitis occurred in all groups, but most commonly in the group of probable disease. For example, Vitreous haze was most frequently seen in our patients with probable sarcoidosis (40.0%), followed by the definite group (37.5%), and the group classified as indefinite sarcoidosis (35.3%). Snowballs as a characteristic finding of intermediate uveitis dominated in our probable group (50.0%).

In terms of posterior involvement, peripheral chorioretinal lesions were particularly common with around 30% in all groups, except in the definite group (11.8%). Other hallmarks of posterior uveitis were the presence of granulomas (3.6%) and papillitis (8.3%), which were predominantly seen in the definite group (Figure 2). More details are presented in Table 4.

#### 3.2.2. Ocular Complications

The most common complications were posterior synechiae (26.2%, 22/84), macular oedema (25%, 21/84), secondary glaucoma (10.7%, 9/84), and cataract (7.1%, 6/84). The distribution of complications across the groups was homogeneous. However, the group of definite patients showed the most significant outlier in their rate of secondary glaucoma at 20.8% (Figure 3).

### 3.3. Laboratory Findings

Laboratory results revealed significantly elevated levels of serum ACE in 34.8% of patients, especially in the definite (40.0%, 4/10) and presumed groups (46.7%, 7/15), while lower in the probable (14.3%, 1/7) and indefinite groups (14.3%, 2/14). This difference was shown to be significant (Wilcoxon rank-sum test *p* = 0.022). The highest average level of serum ACE (83.5 U/l) was detected in the presumed group.

Elevated serum sIL-2R levels were present in 76.2% of patients (32/42). The highest average serum sIL-2R levels were measured in the presumed disease group (1325.2 U/mL). However, there were no significant differences between any of the groups (Kruskal–Wallis test *p* > 0.05). Laboratory values of patients with definite and presumed diagnoses showed a sensitivity of 44.0% for serum ACE and 70.6% for sIL-2R.

Lymphopenia was seen in 32.4% of the patients (12/37). No significant differences were found when comparing all groups (Kruskal-Wallis test *p* > 0.05).

The laboratory values of 19 patients who received ACE inhibitors (*n* = 7) or systemic immunosuppressive therapy (*n* = 14) at first presentation were excluded from the analysis.

## 4. Discussion

Sarcoidosis is a systemic disease, often affecting multiple organs. In many cases, establishing the diagnosis remains challenging. Ocular involvement occurs in up to 79% of cases, with uveitis being the most frequent form in 20–50% [3,13]. To preserve vision and secure early detection of systemic involvement, precise diagnostic tools and criteria are essential. However, there are challenges to deriving a definite diagnosis solely from eye involvement. In only a few cases, histopathology may provide the diagnosis, e.g., due to conjunctival granulomas. Therefore, the combination of clinical signs, laboratory parameters, and further diagnostic imaging is essential. To yield relevant results, we followed the IWOS criteria. This is the first study applying current IWOS criteria to a German population to reevaluate diagnostics for ocular sarcoidosis.

In this study, data from 84 patients whose disease was classified as ocular sarcoidosis by uveitis specialists were analyzed. Age and visual acuity were evenly distributed. The two-tiered age distribution described in previews articles was slightly apparent among the female patients [13]. Reid et al. similarly failed to demonstrate a peak in younger age in their study population in Northern Ireland, reinforcing the relative predominance of older patients in Europe compared to African Americans or Asians [14]. In accordance with our data, the disease is more prevalent in the female population, as other studies have indicated [6,15]. In contrast to any other group, there was a higher ratio of male patients in the indefinite group. Thus, male patients are more likely to exhibit an uncharacteristic, preferential ocular course of the disease.

Systemic sarcoidosis can often be confirmed by means of biopsies and scans of other involved organs, most often from the lung and the skin. This is also reflected in our study, with significantly more frequent lung involvement in the definite and presumed groups compared to the probable and indefinite groups (94.7% vs. 7.4%). Furthermore, patients with a definite diagnosis showed a greater association with skin (41.7% vs. 5%) and nervous system (29.2% vs. 0%) involvement compared to the other groups. Niederer et al. reported an affection rate of the nervous system of 13% in patients with sarcoid uveitis [15]. These data support possible clustering between patients with ocular involvement only, with ocular as well as pulmonary disease, or with further systemic involvement. Schupp et al. have also favored clustering of sarcoidosis patients and postulated, among others, an ocular-cardiac-cutaneous-central nervous system disease involvement group [16]. In 38.3% of our patients with definite or presumed disease, the diagnosis was established based on ocular involvement. Previous studies from Europe demonstrated primary eye involvement in 21.2% to 62.7% [5,6,17]. In 80%, ocular sarcoidosis progresses to second organ involvement at a rate of 14%/person-year [14]. These findings further support the value of early ocular examinations.

In this study, not all patients could be categorized into one of the three—definite, presumed, or probable—IWOS groups. Consequently, an indefinite group remains, although the diagnosis is sometimes clinically plausible with positive chest X-ray findings. Suggesting limited usability in everyday clinical practice, these results support the necessity for further development of the IWOS criteria.

A further subgroup analysis showed a significantly increased share of intermediate uveitis in the group of probable diagnosis as compared to the definite group. Intermediate uveitis has significantly less association with skin involvement than posterior uveitis and thus fewer possibilities for a dermal biopsy. We assume this could be one reason for the higher ratio of probable than definite cases of ocular sarcoidosis in patients with intermediate uveitis. Patients in the definite group were more likely to have posterior uveitis and presented with granulomas. However, all IWOS groups revealed patients with anterior (31.0%), intermediate (32.1%) as well as posterior (17.6%) uveitis. This is in line with previous observations showing a largely homogeneous distribution across the anatomical variants [14,15,18]. However, one German study reported predominantly uveitis anterior in their population with 76.4% [5]. This result may be related to the fact that easier-to-treat cases arrive less at a tertiary center.

Although uveitis is considered a rare disease, it is one of the main causes of blindness in the working-age population [19]. This is mainly related to secondary complications that may occur in any group. At the time of the first visit to our center, 47 out of 84 patients (56.0%) already presented at least one ocular complication. Radosavljević et al. also reported a high rate of complications in ocular sarcoidosis. This was particularly related to the occurrence of secondary glaucoma in 22.7% of sarcoid patients with uveitis [7]. Moreover, Reid et al. described ocular hypertension as a common complication at 36% [14]. These results strongly support the importance of regular intraocular pressure monitoring.

Commonly, the diagnosis of sarcoidosis is supported by laboratory findings. In previous studies, serum ACE has been identified as a meaningful parameter [11]. Previous studies have already suggested that further serological biomarkers may even have a higher value for ocular sarcoidosis. This is particularly true in patients receiving ACE inhibitors to control hypertension, a treatment that is becoming more widespread. This limitation also arises in our cohort. Consequently, the alternative use of sIL-2R as a blood biomarker has risen. While elevated sIL-2R levels were elevated in all of our IWOS subgroups, the highest values were observed in patients with definite and presumed sarcoidosis. Here, the sensitivity reached 70.6%, in contrast to only 44.0% using ACE data. Previous studies also revealed the higher sensitivity of sIL-2R compared to serum ACE levels [20]. This is in line with the known association of ACE with granulomata load, which predominantly occurs extraocular [21]. Thus, its serum level serves as a lagging indicator if sarcoidosis originates in the eye—similar to biopsies and scans outlined above. Therefore, we currently prefer the measurement of sIL-2R, which has not only been shown to serve as an important biomarker for earlier detection of ocular sarcoidosis but has also been recently suggested to be preferred as a diagnostic criterion [22,23].

Limitations of this study result from its retrospective nature, for instance, partially incomplete documentation of clinical findings or the lack of laboratory parameters and imaging. In particular, a biopsy was not performed in all cases. One reason might be the high invasive nature of the examination as well as the lack of a possible site for a biopsy. Regarding laboratory measurements, serum lysozyme was not determined, which has recently been shown to be a useful diagnostic marker for ocular sarcoidosis [24]. Lysozyme levels were further considered as a monitoring parameter for ocular inflammation [25]. With respect to these additional parameters, comparative studies are still warranted, which may focus on the diagnostic and prognostic significance in a differentiated manner. Overall, it should be noted that serum ACE, lysozyme and sIl-2R may also be elevated due to other diseases, such as malignant lymphoma or infections [26,27].

Furthermore, patients presenting in a tertiary center represent a heterogeneous group with a rather high severity of disease. The strength of this study is the structured data collection from a single center on a large number of patients despite it being a relatively rare disease.

## 5. Conclusions

We investigated how the standard IWOS criteria can be applied and extended to allow for a more reliable and earlier diagnosis. Earlier diagnosis promises to prevent complications and thus to preserve vision. Cluster analysis may help to uncover systemic involvement, identify at-risk patients, and develop individualized treatment approaches.

In general, the inability to diagnose sarcoidosis with certainty based on ocular findings remains a challenge. While the disease often originates in the eye, sarcoidosis-specific treatment is delayed until it manifests in other organs. The presented data support the relevance of a thorough ophthalmic workup as well as the importance of interdisciplinary collaboration.

Besides the analysis of clinical parameters and laboratory findings, the further development of imaging techniques might be of high importance in the diagnosis of ocular sarcoidosis. The need for a more precise diagnosis will play a decisive role in the future for specific therapy options in addition to the classic general immunosuppressants.

## Figures and Tables

**Figure 1 cells-11-00089-f001:**
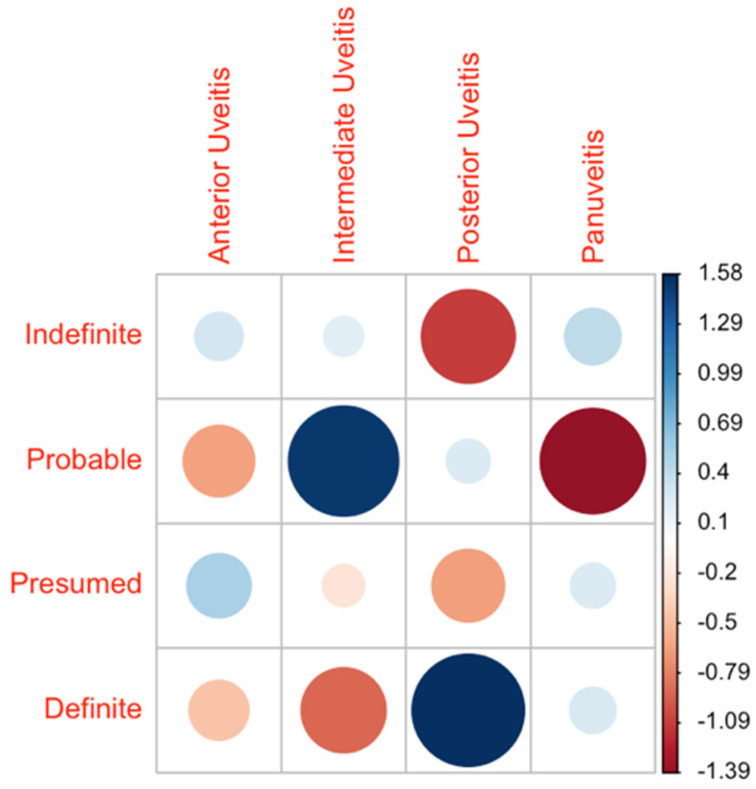
Distribution of uveitis forms across the diagnostic groups. The data are presented as a correlation matrix. The blue color represents a higher-than-expected probability of occurrence, and red represents a lower probability. The larger the circles, the higher the deviation from the expected value given a homogeneous distribution across all groups.

**Figure 2 cells-11-00089-f002:**
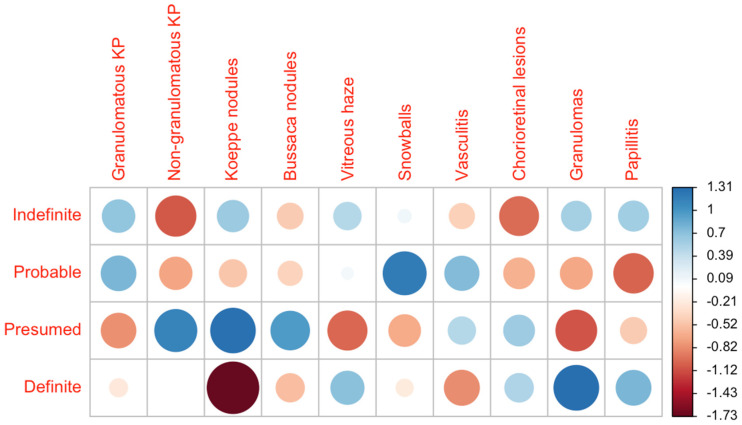
Distribution of the single ocular manifestations across the diagnostic groups. The data are presented as a correlation matrix. The blue color represents a higher-than-expected probability of occurrence, and red represents a lower probability. KP = keratic precipitates. The larger the circles, the higher the deviation from the expected value given a homogeneous distribution across all groups.

**Figure 3 cells-11-00089-f003:**
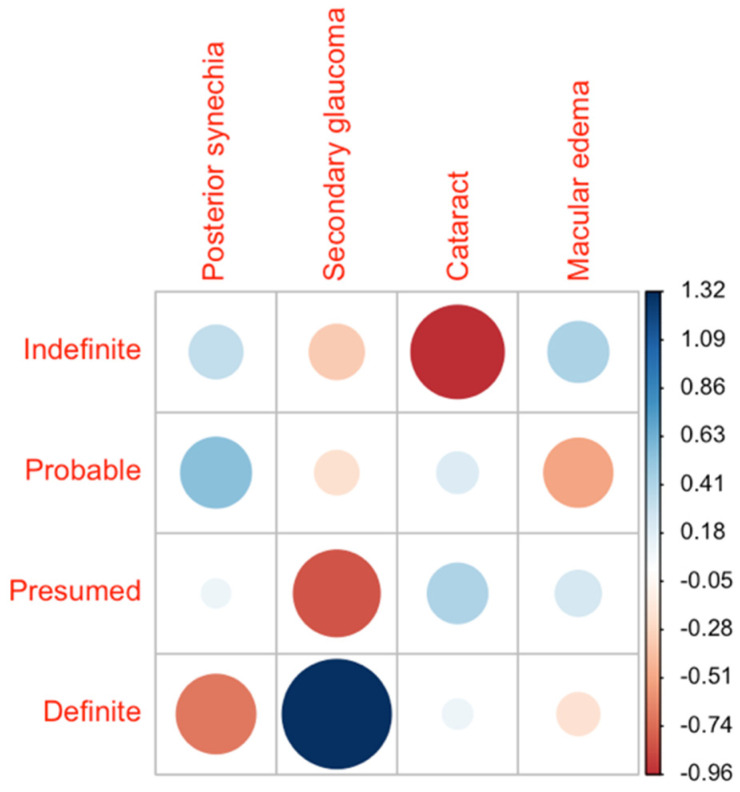
Distribution of the single ocular complications across the diagnostic groups. The data are presented as a correlation matrix. The blue color represents a higher-than-expected probability of occurrence, and red represents a lower probability. The larger the circles, the higher the deviation from the expected value given a homogeneous distribution across all groups.

**Table 1 cells-11-00089-t001:** International Workshop on Ocular Sarcoidosis criteria (Adapted from [11]). The following key findings are used to determine the diagnosis of ocular sarcoidosis. They are listed as: (**a**) Intraocular findings and (**b**) Further diagnostic criteria.

**Intraocular findings**
1. Granulomatous keratic precipitates, iris nodules/granulomas (Koeppe/Busacca)
2. Nodules in the trabecular meshwork (TM) and/or tent-like peripheral anterior synechiae
3. Vitreous opacities “snowballs”
4. Peripheral chorioretinal lesions (active and/or atrophic)
5. Segmental periphlebitis (±“candle wax phenomenon”) and/or retinal macro aneurysm in an inflamed eye
6. Optic nerve granulomas and/or solitary choroidal granulomas
7. Mutuality
(**a**)
**Further diagnostic criteria**
1. Chest x-ray or computer tomography findings with bilateral hilar lymphadenopathy (BHL)
2. Negative tuberculin skin test or interferon-gamma releasing assay
3. Elevated serum Angiotensin-converting enzyme (ACE) values
4. Elevated serum lysozyme values
5. CD4/CD8 ratio > 3.5 in bronchoalveolar lavage (BAL)
6. Positive gallium-67 scintigraphy or 18F-fluorodeoxyglucose positron emission tomography (PET) imaging
7. Lymphopenia (<1000 cells/µL)
8. Parenchymal lung changes in line with sarcoidosis
9. Elevated soluble interleukin-2 receptor (sIL-2R) ^1^
(**b**)

^1^ Adjusted criteria.

**Table 2 cells-11-00089-t002:** Based on the International Workshop on Ocular Sarcoidosis diagnostic criteria, 4 levels of evidence for ocular sarcoidosis were derived (Adapted from [11]).

Level of Evidence	Diagnostic Criteria
Definite ocular sarcoidosis	Biopsy-confirmed diagnosis with clinically corresponding uveitis
Presumed ocular sarcoidosis	Biopsy was not performed or negative; however, chest X-ray/computer tomography findings with hilum changes and 2 positive listed intraocular findings
Probable ocular sarcoidosis	Biopsy was not performed or negative; no positive chest X-ray/computer tomography findings, but 3 of the listed intraocular findings and 2 positive laboratory/imaging findings are available
Indefinite ocular sarcoidosis	Ocular sarcoidosis suspected intraocular findings or investigations without fitting into a pattern of the above categories

**Table 3 cells-11-00089-t003:** Patient characteristics and sarcoid manifestation.

		Definite	Presumed	Probable	Indefinite	Total
Count *n* (%)		24 (28.6)	33 (39.3)	10 (11.9)	17 (20.2)	84 (100)
Gender *n* (%)	Female	17 (70.8)	20 (60.6)	6 (60.0)	7 (41.2)	50 (59.5)
	Male	7 (29.1)	13 (39.4)	4 (40.0)	10 (58.8)	34 (40.5)
Age mean (Range)		52 (31–77)	58 (28–78)	51 (11–87)	45 (8–87)	53 (8–87)
Extraocular Manifestations *n* (%)	Lung	21 (87.5)	33 (100.0)	0 (0.0)	2 (11.8)	56 (66.7)
	Skin	10 (41.7)	3 (9.1)	0 (0.0)	0 (0.0)	13 (15.5)
	Lymph nodes	3 (12.5)	1 (3.0)	0 (0.0)	0 (0.0)	4 (4.8)
	Nervous system	7 (29.2)	0 (0.0)	0 (0.0)	0 (0.0)	7 (8.3)
	Heart	3 (12.5)	1 (3.0)	0 (0.0)	0 (0.0)	4 (4.8)
	Liver	2 (8.3)	1 (3.0)	0 (0.0)	0 (0.0)	3 (3.6)
	Spleen	5 (20.8)	1 (3.0)	0 (0.0)	0 (0.0)	6 (7.1)
	Kidney	0 (0.0)	0 (0.0)	0 (0.0)	0 (0.0)	0 (0.0)
	Joints	1 (4.2)	2 (6.1)	0 (0.0)	1 (5.9)	4 (4.8)

**Table 4 cells-11-00089-t004:** Clinical manifestations and complications of ocular sarcoidosis patients.

		Definite	Presumed	Probable	Indefinite	Total
	Number of patients	24	33	10	17	84
	Bilaterality	79.2% (19/24)	70.0% (23/33)	100.0% (10/10)	58.8% (10/17)	73.8% (62/84)
Anterior	Total	25.0% (6/24)	36.4% (12/33)	20.0% (2/10)	35.3% (6/17)	31.0% (26/84)
	Granulomatous keratic precipitates (KP)	25% (6/24)	21.2% (7/33)	50.0% (5/10)	35.3% (6/17)	28.6% (24/84)
	Non-granulomatous KP	25.0% (6/24)	36.4% (12/33)	20.0% (2/10)	11.8% (2/17)	26.2% (22/84)
	Koeppe nodules	0.0% (0/24)	21.2% (7/33)	10.0% (1/10)	17.6% (3/17)	13.1% (11/84)
	Bussaca nodules	0.0% (0/24)	3.0% (1/33)	0.0% (0/10)	29.4% (0/17)	1.2% (1/84)
Intermediate	Total	20.8% (5/24)	30.3% (10/33)	60.0% (6/10)	35.3% (6/17)	32.1% (27/84)
	Vitreous haze	37.5% (9/24)	21.2% (7/33)	40.0% (4/10)	35.3% (6/17)	31.0% (26/84)
	Snowballs	20.8% (5/24)	18.2% (6/33)	50.0% (5/10)	23.5% (4/17)	23.8% (20/84)
	Vasculitis	4.2% (1/24)	12.1% (4/33)	20.0% (2/10)	5.9% (1/17)	9.5% (8/84)
Posterior	Total	29.2% (7/24)	12.1% (4/33)	20.0% (2/10)	5.9% (1/17)	17.6% (14/84)
	Chorioretinal lesions	29.2% (7/24)	30.3% (10/33)	20.0% (2/10)	11.8% (2/17)	25% (21/84)
	Granulomas	8.3% (2/24)	0.0% (0/33)	0.0% (0/10)	5.9% (1/17)	3.6% (3/84)
	Papillitis	12.5% (3/24)	6.1% (2/33)	0.0% (0/10)	11.8% (2/17)	8.3% (7/84)
Panuveitis	Total	20.8% (5/24)	21.2% (7/33)	0.0% (0/10)	23.5% (4/17)	19.0% (16/84)
Complications	Total	62.5% (15/24)	54.5% (18/33)	60.0% (6/10)	47.0% (8/17)	56.0% (47/84)
	Posterior synechia	20.8% (5/24)	27.3% (9/33)	40.0% (4/10)	23.5% (4/17)	26.2% (22/84)
	Secondary glaucoma	20.8% (5/24)	6.1% (2/33)	10.0% (1/10)	5.9% (1/17)	10.7% (9/84)
	Cataract	8.3% (2/24)	9.1% (3/33)	10.0% (1/10)	0% (0/17)	7.1% (6/84)
	Macular edema	25.0% (6/24)	27.3% (9/33)	20.0% (2/10)	23.5% (4/17)	25% (21/84)

## Data Availability

The data that support the findings of this study are available from the corresponding author, L.S.z.B., upon reasonable request.

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
