# Peer review of "Findings and Graduation of Sarcoidosis-Related Uveitis: A Single-Center Study"

_cells, 2021, doi:10.3390/cells11010089_

Round 1
Reviewer 1 Report
This is an interesting study applying the IWOS criteria for sarcoidosis. Although it is interesting to perform correlation matrix data presentation, the relevance appears limited. Otherwise the article is well-written and gives clear information
A major drawback is the complete absence of serum lysozyme as a laboratory test. The authors should explain in the discussion why they completely scotomised this parameter and mentione it as a significant test, much more useful than ACE. To support this argument they can cite: Papasavvas et al. The Comparative Value of Serum Angiotensin Converting Enzyme (ACE) and Lysozyme and the Use of Polyclonal Antibody Activation in the Work-up of Ocular Sarcoidosis. Diagnostics (Basel). 2021 Mar 29;11(4):608. doi: 10.3390/diagnostics11040608. PMID: 33805490; PMCID: PMC8066732.
Author Response
Response to Reviewer 1 Comments
Point 1: A major drawback is the complete absence of serum lysozyme as a laboratory test. The authors should explain in the discussion why they completely scotomised this parameter and mentione it as a significant test, much more useful than ACE. To support this argument, they can cite: Papasavvas et al. The Comparative Value of Serum Angiotensin Converting Enzyme (ACE) and Lysozyme and the Use of Polyclonal Antibody Activation in the Work-up of Ocular Sarcoidosis. Diagnostics (Basel). 2021 Mar 29;11(4):608. doi: 10.3390/diagnostics11040608. PMID: 33805490; PMCID: PMC8066732.

Response 1: This point is well taken. Lysozyme is indeed an independent parameter included in the IWOS criteria (Mochizuki M, Smith JR, Takase H, et al. Br J Ophthalmol 2019; 103: 1418–1422). However, the criteria imply that it is not necessary to perform all systemic examinations to establish the diagnosis. For a probable diagnosis, two out of seven selected systemic investigations are sufficient. This further appears to be economically reasonable. In our study, we chose to measure serum ACE levels, which is a well-established marker. The advantage arises from the circumstance that lysozyme may also be elevated in other diseases such as syphilis or tuberculosis, which are differential diagnoses of high relevance in the field of uveitis. Therefore, lysozyme is rather recommended as a monitoring parameter (Sahin, 2016; Tomita, 1999). In our mind, it is important to emphasize that the soluble interleukin-2 receptor seems to have an even higher diagnostic value than serum ACE for ocular sarcoidosis. This is in agreement with previous observations (Groen-Hakan, 2017). Unfortunately, the comparison with the soluble interleukin-2 receptor is not included in the mentioned work (Papasavvas, 2021). Taken together, this shows that further comparative work can be useful.
As per your suggestion, we have added the following reference (Papasavvas, 2021). In order to provide a broad view on this aspect, the following additional references were also cited (Groen-Hakan, 2017; Sahin, 2016; Tomita, 1999).
We have listed these aspects in the discussion (line 287) and underlined them accordingly with literature:
“Regarding laboratory measurements, serum lysozyme was not determined, which has recently been shown to be a useful marker for ocular sarcoidosis [24]. However, its diagnostic value is reduced by the possibility of increased levels in other diseases such as syphilis or tuberculosis, both of which are important differential diagnoses in the field of uveitis [25, 26]. With respect to these further parameters, comparative studies may still be indicated, which can be focused on the diagnostic and prognostic significance in a differentiated manner.”
- Groen-Hakan, F.; Eurelings, L.; ten Berge, J. C.; van Laar, J.; Ramakers, C.; Dik, W. A.; Rothova, A. (2017). Diagnostic Value of Serum-Soluble Interleukin 2 Receptor Levels vs Angiotensin-Converting Enzyme in Patients With Sarcoidosis-Associated Uveitis. JAMA Ophthalmol. 2017, 135, 1352–1358. https://doi.org/10.1001/jamaophthalmol.2017.4771
- Papasavvas, I.; Gehrig, B.; Herbort, C. P., Jr. The Comparative Value of Serum Angiotensin Converting Enzyme (ACE) and Lysozyme and the Use of Polyclonal Antibody Activation in the Work-up of Ocular Sarcoidosis. Diagnostics 2021, 11, 608. https://doi.org/10.3390/diagnostics11040608
- Sahin, O.; Ziaei, A.; KaraismailoÄŸlu, E.; Taheri, N. The serum angiotensin converting enzyme and lysozyme levels in patients with ocular involvement of autoimmune and infectious diseases. BMC Ophthalmol. 2016, 16, 19. https://doi.org/10.1186/s12886-016-0194-4
- Tomita, H.; Sato, S.; Matsuda, R.; Sugiura, Y.; Kawaguchi, H.; Niimi, T.; Yoshida, S.; Morishita, M. Serum lysozyme levels and clinical features of sarcoidosis. Lung 1999, 177, 161–167. https://doi.org/10.1007/pl00007637
Reviewer 2 Report
Sarcoidosis is one of the most frequently detected causes of uveitis in tetrtiary centers among patients previously defined as idiopathic. Delay in diagnosis is very common. As Authors mention - there are only few reports on the application of IWOS guidelines for ocular sarcoidosis.
The work is well-written, coherent and facilitates the review of diagnostic criteria in accordance with the division created by IWOS experts along with their practical application.
It would be interesting to know the data on the diagnosis and treatment before referral to the tertiary center. Perhaps supplementing the work with data on the types of treatment implemented depending on the group according to IWOS would be valuable, too. It could also be a correlation matrix. The percentage of patients using ACE inhibitors in the study group is missing, which, as the Authors themselves write, is an important feature. Have there been any changes in the types of laboratory tests performed over time (14 years)? Was the conduct of specific laboratory tests similar in the entire group?
Autjors investigated how the standard IWOS criteria can be applied and extended to allow for a more reliable and earlier diagnosis. However, it is difficult to be fully convinced that the work confirms this belief. Sarcoidosis often remains a difficult disease to diagnose regardless of trying to systematize the criteria - which is very necessary, but still not quite sufficient for an early diagnosis.
Author Response
Response to Reviewer 2 Comments
Point 1: It would be interesting to know the data on the diagnosis and treatment before referral to the tertiary center. Perhaps supplementing the work with data on the types of treatment implemented depending on the group according to IWOS would be valuable, too. It could also be a correlation matrix.
Response 1: Thank you for this notion. We agree on the fact that therapeutic management of ocular sarcoidosis is an important issue, which still requires major scientific attention. However, in this study we have deliberately chosen to focus on diagnostics exclusively. Certainly, the influence of laboratory parameters by systemic immunosuppressive therapy is of great relevance. As mentioned in the methods section, laboratory values of patients who initially already presented with systemic therapy (n=14) were excluded from the analysis. Regarding your comment, we will add this number to the results section explicitly (line 196):
“The laboratory values of 19 patients who received ACE inhibitors (n=7) or systemic immunosuppressive therapy (n=14) at first presentation were excluded from the analysis.”
Point 2: The percentage of patients using ACE inhibitors in the study group is missing, which, as the Authors themselves write, is an important feature.
Response 2: This point is well taken. We have addressed this aspect accordingly and the number of patients (n=7) was added in the results section (line=196; see point 1). As mentioned in the methods section, laboratory values of patients taking ACE inhibitors were excluded from the analysis.
Point 3: Have there been any changes in the types of laboratory tests performed over time (14 years)? Was the conduct of specific laboratory tests similar in the entire group?
Response 3: Over time, there were no changes in the laboratory tests performed. From the beginning, sIL-2R was also regularly determined. Only the laboratory devices have changed over time. In this respect, the individual laboratory values were normalized with respect to the norm ranges of the respective device for comparison, as described in the method section.
Point 4: Authors investigated how the standard IWOS criteria can be applied and extended to allow for a more reliable and earlier diagnosis. However, it is difficult to be fully convinced that the work confirms this belief. Sarcoidosis often remains a difficult disease to diagnose regardless of trying to systematize the criteria - which is very necessary, but still not quite sufficient for an early diagnosis.
Response 4:
Indeed, further diagnostics would be helpful to confirm the diagnosis more reliably. However, we believe that a classification and thus better understanding of the clinical signs as well as laboratory values will enhance the understanding of the disease in a broader perspective and help to identify ocular sarcoidosis earlier among other differential diagnoses.
Round 2
Reviewer 1 Report
The response of the authors concerning the laboratory test lysozyme are unsatisfactory, unacceptable and misleading. The fact that a test can be positive in other diseases is not a reason not to perform a test. This is the case for most tests.
The way the authors respond to this remark indicates that the test is not useful, which is absolitely not the case. This test is cited as essential in the IWOS criteria. The authors can respond that it is not in their habit to perform this test but it is not scientically/ medically justified to downplay the importance of this test.
This part of the text has to be rephrased to maintain the place of lysozyme as an important test.
Author Response
Response 2 to Reviewer 1 Comments
Point 1: The response of the authors concerning the laboratory test lysozyme are unsatisfactory, unacceptable and misleading. The fact that a test can be positive in other diseases is not a reason not to perform a test. This is the case for most tests.
The way the authors respond to this remark indicates that the test is not useful, which is absolitely not the case. This test is cited as essential in the IWOS criteria. The authors can respond that it is not in their habit to perform this test but it is not scientically/ medically justified to downplay the importance of this test.
This part of the text has to be rephrased to maintain the place of lysozyme as an important test.
Response 1:
We apologize for the misleading comment. We did not wish to suggest that lysozyme is not a useful parameter per se and have rephrased the paragraph on lysozyme:
“Regarding laboratory measurements, serum lysozyme was not determined, which has recently been shown to be a useful diagnostic marker for ocular sarcoidosis [24]. Lysozyme levels were further considered as a monitoring parameter for ocular inflammation [25]. With respect to these additional parameters, comparative studies are still warranted, which may focus on the diagnostic and prognostic significance in a differentiated manner.
Overall, it should be noted that serum ACE, lysozyme and sIl-2R may also be elevated due to other diseases, such as malignant lymphoma or infections [26, 27].”
- Papasavvas, I.; Gehrig, B.; Herbort, C. P., Jr. The Comparative Value of Serum Angiotensin Converting Enzyme (ACE) and Lysozyme and the Use of Polyclonal Antibody Activation in the Work-up of Ocular Sarcoidosis. Diagnostics 2021, 11, 608. https://doi.org/10.3390/diagnostics11040608
- Tomita, H.; Sato, S.; Matsuda, R.; Sugiura, Y.; Kawaguchi, H.; Niimi, T.; Yoshida, S.; Morishita, M. Serum lysozyme levels and clinical features of sarcoidosis. Lung 1999, 177, 161–167. https://doi.org/10.1007/pl00007637
- Sahin, O.; Ziaei, A.; KaraismailoÄŸlu, E.; Taheri, N. The serum angiotensin converting enzyme and lysozyme levels in patients with ocular involvement of autoimmune and infectious diseases. BMC Ophthalmol. 2016, 16, 19. https://doi.org/10.1186/s12886-016-0194-4
- Murakami, K.; Koh, J.; Taruya, J.; Ito, H. Mimicking the Recurrence of Malignant Lymphoma. Case Rep Neurol. 2021, 13, 605–612. https://doi.org/10.1159/000518378
Round 3
Reviewer 1 Report
The rephrased lines 287-293 are now more appropriate now more appropriate